# Translation, validation and psychometric evaluation of the Persian (Farsi) version of the Low Anterior Resection Syndrome Score (LARS-P)

**Mohammad Reza Keramati**[1,2☯]*, **Ali Abbaszadeh-Kasbi**[1,2☯], **Amir Keshvari**[1,2], **Seyed Mohsen Ahmadi-Tafti**[1,2], **Behnam Behboudi**[1,2], **Alireza Kazemeini**[1,2], **Mohammad Sadegh Fazeli**[1,2]

1 Division of Colorectal Surgery, Department of Surgery, Tehran University of Medical Sciences, Tehran, Iran, 2 Colorectal Surgery Research Center, Tehran University of Medical Sciences, Tehran, Iran

☯ These authors contributed equally to this work.
* mr-keramati@tums.ac.ir

## Abstract

### Introduction

Low anterior resection (LAR) for rectal cancer affects bowel function after the operation, causing a group of symptoms known as LAR Syndrome (LARS). LARS score is a patient-reported questionnaire to assess bowel dysfunction after the LAR operation. This study performed to validate the Persian (Farsi) translation of the LARS score and to investigate the psychometric properties of the score. The impact of LARS on the Quality of Life (QoL) of patients was also assessed.

### Materials and methods

The LARS score was translated into Persian. Participants with a history of rectal cancer and low anterior resection were asked to complete the LARS score questionnaire. They were also asked a single question evaluating the impact of bowel function on QoL. Discriminative validity, convergent validity, sensitivity, and specificity of the questionnaire were calculated. A group of patients completed the score twice to assess the reliability of the questionnaire.

### Results

From 358 patients with rectal cancer, 101 participants completed the Persian questionnaire. Answers of a high fraction of participants showed a moderate/perfect fit between their LARS score and their QoL. The Persian score demonstrated good convergent validity. It was able to differentiate between participants in terms of gender and T staging of the primary tumor. The score had high reliability.

**Data Availability Statement:** All relevant data are within the paper and its Supporting Information file.

**Funding:** The author(s) received no specific funding for this work.

**Competing interests:** The authors have declared that no competing interests exist.

## Conclusion

The Persian translation of the LARS score has excellent psychometric properties compared to previous translations in other languages. Therefore, it is a valid and reliable questionnaire to assess LARS.

## 1. Introduction

Colorectal cancer is one of the most common malignancies in humans. It ranks third for incidence and second for mortality, among other malignancies [1]. Adoption of proper practices in early diagnosis and treatment of colorectal cancers has led to a decline in mortality [1, 2]. Laparoscopic or open Total Mesorectal Excision (TME) is currently the standard and potentially curative surgical approach for rectal cancers [3, 4]. Although an oncologic cure is the principal goal of surgical treatment of rectal cancer, intestinal continuity and preservation of sphincter function are valuable goals [5]; therefore, Sphincter Preserving Surgeries, i.e., Low Anterior Resection (LAR) and Ultra LAR are becoming more popular among surgeons [6].

Following sphincter preserving surgeries, a large group of patients develops some changes in bowel habits, and a group of defecatory symptoms known as Low Anterior Resection Syndrome (LARS). Symptoms of LARS vary from constipation and obstructed defecation to daily incontinence episodes. The wide range of LARS symptoms makes the development of a definition or classification difficult [7]. Several instruments and questionnaires have been developed to measure the bowel function that can be used after these surgical operations [8–13].

The LARS score is one of the most common questionnaires used to assess functional outcomes and symptoms after LAR surgical operation for rectal cancer. In 2012, this score was initially designed and validated in Danish by Emmertsen et al. [13]. Validity and reliability of translated versions of this score have also been assessed in other languages, including English [14], Chinese [15], Lithuanian [16], Dutch [17], Swedish, Spanish, German [18], and Japanese [19]. The aim of the present study was to translate the LARS score into Persian (Farsi) language and to test its psychometric properties.

## 2. Materials and methods

### 2.1 LARS score

The LARS score contains five items. The score-values were based on the item's impact on the Quality of Life (QoL) of the patients. Except for the third item (frequency of bowel movements), which has four options, all the other items have three choices. The total score is ranging from 0 to 42. Scores of 0 to 20 represent "no LARS"; 21 to 29 as "minor LARS", and 30–42 as "major LARS" [13, 14].

### 2.2 Translation

First, we acquired permission from the original authors. Then, two Persian (Farsi) speaking researchers who were fluent in both Persian and English translated the questionnaire to Persian independently. The translators tested the translations for any discrepancy to reach a final consensus and prepare the provisional Persian version. Then, the Persian version was translated back into English by another native English-speaking colleague who was also fluent in the Persian language. The English back-translated version was then compared with the original version to make sure there are no conceptual discrepancies [20].

## 2.3 Participants and data gathering

After approval by the ethics committee of Tehran University of Medical Sciences, the participants were recruited from Imam Hospital, a tertiary referral university-affiliated hospital. Participants were included if they were eighteen years or older, confirmed histologic diagnosis of rectal cancer, history of curative sphincter preserving rectal resection surgery and diverting stoma reversal. Exclusion criteria were history of palliative surgery for rectal cancer, recurrence or metastasis from primary cancer, receiving adjuvant therapy, other bowel or anus disorders (including irritable bowel syndrome, inflammatory bowel disease, and known anal sphincter dysfunction), cognition or language problems.

In accordance with the inclusion and exclusion criteria and from the patients referred to the division of colorectal surgery from February 2018 to February 2020, eligible patients were invited to join the study. The participants can be considered representative of a larger population as they had been referred from the whole country. The participants were informed about the aim of the study during a clinic visit, informed consent was taken, and they were asked to fill out the questionnaire. The completed questionnaires were collected and then thoroughly evaluated for errors.

## 2.4 Statistical analysis

SPSS software version 25 (IBM SPSS Statistics for Windows, Armonk, NY: IBM Corp) was utilized for statistical analysis. P-values less than 0.05 were regarded as statistically significant. Categorical data will be shown as number (percent), and continuous data as mean ± standard deviation in this paper. In below, other parts of the analysis will be described comprehensively.

## 2.5 Content validity and face validity

Content validity of the translated questionnaire was assessed by a panel of experts including 20 university faculty members using a 3-point Likert scale (poor, average and good). Each expert judged the items of the questionnaire independently. The Persian version of the questionnaire received a good content validity from the experts. To assess the face validity of the translated questionnaire, a total of 100 patients were asked to verify the degree of comprehension and adequacy of the questions. The translated questionnaire was easy to understand and acceptable for all of the patients.

## 2.6 Convergent validity

To analyze the convergent validity and according to previously validated questionnaires, one question on QoL was included in the translated version of the LARS score questionnaire [14]. Convergent validity shows if two measurements that are supposed to have the same construct are related or not. Persian translation of the question "Overall, how much does your bowel function affect your quality of life?" was added to the end of the translated questionnaire. Correlation of the LARS score against the QoL question was evaluated to test the potential statistical relationship between the LARS score and the QoL of the patients. As mentioned above, the LARS score was categorized into no, minor, or major LARS. Based on the influence of LARS on the quality of life, patients were also classified into three QoL groups, including no/mild, moderate, or severe impact on QoL. A possible correlation between the LARS score group and the QoL group was assessed. "Perfect fit" meant both groups (LARS score and QoL) are matched completely. An incompatibility in one category was considered as "moderate fit" and a total mismatch as "no fit". Moreover, the sensitivity and specificity of the translated score for identifying patients with a moderate to severe impact on QoL were also calculated.

## 2.7 Discriminative validity

To evaluate the discriminative validity of the questionnaire, we tested the ability of the translated version of the LARS score to make a distinction between various groups of patients who were expected to differ with regards to LARS [15, 17, 19]. This included age, gender, distance of the tumor from the anal verge, T and N staging of the tumor, total TNM staging of the tumor, extent of mesorectal excision (total versus partial), and history of neoadjuvant chemoradiation.

## 2.8 Reliability

To assess the reliability of the LARS score, a test-retest analysis was performed. All participants filled out the Persian version of the LARS questionnaire again at least one week (and a maximum of 6 weeks) after they first had completed it [17, 19]. The correlation between the total scores of the questionnaire at the two evaluations was assessed using the Interclass Correlation Coefficient (ICC), Standard Error of Mean (SEM) and Coefficient of Repeatability (CR). Furthermore, for each question of the score, the two responses were compared. A "perfect agreement" meant selecting the same response at two tests. "Moderate agreement" was given when responses at two assessments were different by only one category. "No agreement" showed a difference of two or more classes at the two tests.

# 3. Results

## 3.1 Participants and the LARS score

A total of 358 patients were initially approached. Of those, 104 eligible patients were accepted to join in the study and completed the first questionnaire. Three participants returned incomplete questionnaires, and 101 participants included in the study finally. The median age of the responders was 57 years (range: 32–84 years, mean: 57.2±11.6). Male to female ratio was 1.34, including 58 (57.4%) men and 43 (42.6%) women. For the test-retest part of the study (reliability analysis), from the 45 patients who were asked to fill the LARS score for the second time, 37 patients responded. (Fig 1) Completion rates for the first and second parts of the study were 97.1% and 82.2%, respectively.

All patients had undergone Low Anterior Resection (LAR) and diverting loop ileostomy for their rectal cancer. Stomas were closed at least one month before the assessment of the LARS score. The median time from stoma closure to assessing the LARS score was 77 days (range: 33–139 days, mean: 80.1±28.3). The median time from the LAR operation to evaluate the LARS score was 327 days (range: 126–440 days, mean: 317.1±59.1). The demographic and clinical features of the participants are represented in Table 1.

According to the score, no, minor, and major LARS were detected in 30 (29.7%), 27 (26.7%), and 44(43.6%) participants, respectively. The mean LARS score in the participants reporting no, minor, and major LARS was 13.1±6.5, 24.7±2.7, and 36.5±3.8, respectively. Considering the QoL group, mean LARS scores were 18±5.4, 26.7±5.9, and 35.9±6.6 in no/mild, moderate, and severe groups of QoL.

## 3.2 Convergent validity, sensitivity, and specificity

The LARS groups (no, minor, and major LARS) were compared with the QoL groups (no/mild, moderate, and severe influence on the QoL) to examine convergent validity. The percentage of participants with a perfect fit between the QoL group and the LARS group was 82.1%. A moderate fit was found in 16.9% and no fit in 1% of the participants. It showed that a higher LARS score was associated with a severe influence of symptoms on the QoL of the participants (Table 2).

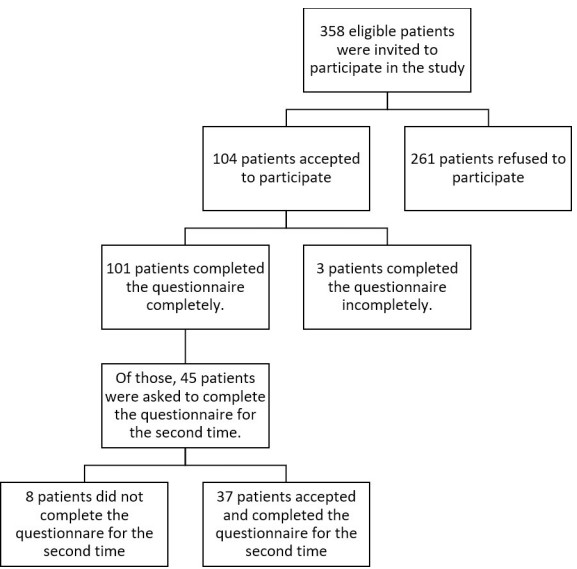

**Fig 1. Flowchart for the patient selection.**

The sensitivity and specificity of the LARS score in identifying patients with moderate and severe impact on QoL were 97.3% and 78.7%, respectively. This shows that the LARS score can predict the influence of symptoms related to LARS on QoL. The ROC curve of the LARS scores predicting patients with moderate and severe influence on QoL revealed an area under the curve of 0.933 (Fig 2).

### 3.3 Discriminative validity

According to the discriminative validity analysis, the LARS score was able to differentiate between male and female participants (p-value = 0.017) and T staging of the primary tumor

**Table 1. Demographic and clinical characteristics of the participants.**

| Variable | | Value |
|---|---|---|
| Age | | 57 (57.2±11.6) |
| Sex | Female | 43 (42.6%) |
| | Male | 58 (57.4%) |
| Stage of Tumor (Union for International Cancer Control TNM Staging) | I | 12 (11.9%) |
| | II | 26 (25.7) |
| | III | 63 (62.4) |
| Extent of mesorectal excision | Total Mesorectal Excision (TME) | 64 (63.4%) |
| | Partial Mesorectal Excision (PME) | 37 (36.6%) |
| History of neoadjuvant chemoradiation | Yes | 89 (88.1%) |
| | No | 12 (11.9%) |
| Tumor distance from the anal verge | | 8 (8.26±2.4) |
| Days from surgery to LARS Score assessment | | 327 (317.1 ±59.1) |
| Days from stoma closure to LARS Score assessment | | 77 (80.1±28.3) |

Categorical data are presented as number (percent), and continuous data as median (mean ± standard deviation).

**Table 2. Convergent validity.**

| | | Impact on the Quality of Life | | |
|---|---|---|---|---|
| | | No | Moderate | Severe |
| LARS Score | No LARS | 26 (25.7%) * | 3 (3.0%) ** | 1 (1.0%) *** |
| | Minor LARS | 7 (6.9%) ** | 20 (19.8%) * | 0 (0.0%) ** |
| | Major LARS | 0 (0.0%) *** | 7 (6.9%) ** | 37 (36.6%) * |

* Perfect fit (82.1%).

** Moderate fit (16.9%).

*** No fit (1%).

Data are presented as "number (percent)".

(p-value = 0.02). However, it was not able to classify participants in terms of their age, the distance of the tumor from the anal verge, lymph node involvement of the tumor, stage of the tumor, extent of mesorectal excision and history of neoadjuvant chemoradiation. Table 3 demonstrates the variables included in the discriminative validity of the score and the results of the analysis.

### 3.4 Test-retest reliability

From the 45 invited participants to complete the questionnaire for the second time, 37 responded (82.2%). The median time between the two tests was 19 days (range: 6–42

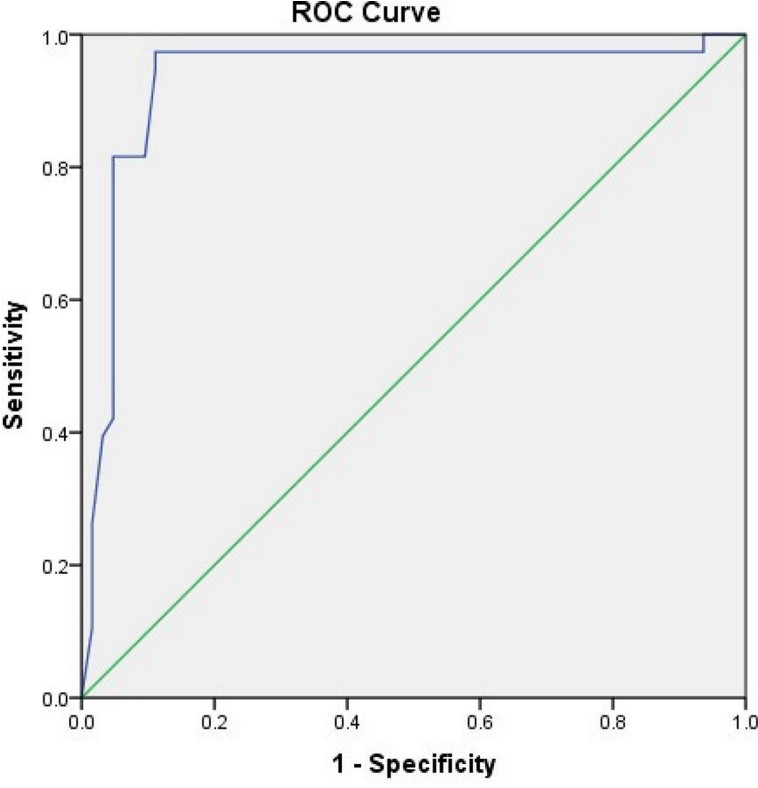

**Fig 2. ROC curve shows the relation between the LARS score and the moderate/severe impact on QoL.**

**Table 3. Discriminative validity of the LARS score.**

| Variable | | P-value |
|---|---|---|
| Age | < 57 years | 0.46 |
| | > = 57 year | |
| Sex | Female | .017* |
| | Male | |
| Distance of tumor from anal verge | <8 | 0.22 |
| | >8 | |
| T staging of tumor (TNM Staging) | T1 | .02* |
| | T2 | |
| | T3 | |
| Lymph node involvement (TNM Staging) | N0 | .23 |
| | N1 | |
| | N2 | |
| Stage of tumor (Union for International Cancer Control TNM Staging) | I | .58 |
| | II | |
| | III | |
| Extent of mesorectal excision | Total Mesorectal Excision (TME) | .89 |
| | Partial Mesorectal Excision (PME) | |
| History of neoadjuvant chemoradiation | Yes | .57 |
| | No | |

* Statistically significant (p-value<0.05).

days). No statistically meaningful difference was found between the LARS scores on the first and second tests (p-value>0.05). Twenty-nine (78.3%) patients stayed in the same LARS group in both tests (perfect agreement), including 9, 7, and 13 patients in the no, minor and major LARS groups, respectively. LARS group of eight (21.7%) participants changed by one category (moderate agreement). (Table 4) Interclass Correlation Coefficient (ICC) was 0.976 that showed an excellent agreement between the test and the retest. Standard error of mean (SEM) and coefficient of repeatability (CR) were 0.154 and 0.429, respectively. Fig 3 depicts the Bland–Altman plot of the differences between LARS scores on the first and second tests.

## 4. Discussion

In this study, the LARS (Low Anterior Rectal Syndrome) score was translated into Persian language and with the help of 101 participants with a history of rectal cancer and low anterior resection, it showed excellent psychometric properties compared to previous translations in other languages. According to this study, this translated questionnaire is valid and reliable to assess LARS in these patients.

In the face of recent advances in the treatment of colorectal cancers, radical surgery is still the backbone of curative treatment for resectable rectal cancer [21]. Abdominoperineal resection and sphincter-preserving operations, including Low Anterior Resection (LAR), are the principal surgical options for resectable rectal cancer. With the introduction of advanced surgical staplers and neoadjuvant treatments, indications for LAR (or Ultra/Very LAR) operation expanded for low rectal cancers. Furthermore, preservation of the sphincter is often as essential as curing rectal cancer for most of the patients. Considering the outcome and

**Table 4. Test-retest reliability.**

| | | LARS Score of the second test (retest) | | |
|---|---|---|---|---|
| | | no LARS | Minor LARS | Major LARS |
| LARS Score of the first test | No LARS | 9 (24.3%) * | 6 (16.2%) ** | 0 (0.0%) *** |
| | Minor LARS | 0 (0.0%) ** | 7 (18.9%) * | 2 (5.4%) ** |
| | Major LARS | 0 (0.0%) *** | 0 (0.0%) ** | 13 (35.1%) * |

\* Perfect agreement.

\*\* Moderate agreement.

\*\*\* No agreement.

Data are presented as "number (percent)".

complications of the surgery, surgeons need to decide if a patient with rectal cancer is a candidate for a sphincter preserving operation or not [22].

## 4.1 Low Anterior Rectal Syndrome (LARS)

Sphincter preservation techniques sustain intestinal continuity to avoid permanent stoma and are considered as the preferred option for patients. However, these may not guarantee satisfactory defecation, continence, and quality of life after the operation [15]. The decreased neo-rectal capacity, elevated intraluminal pressure along with a weakened sphincter after the operation, modify anorectal physiology [23]. A significant number of patients undergoing a LAR will develop a group of symptoms named as the low anterior resection syndrome (LARS). The LARS score is a five-item questionnaire for assessing bowel function after LAR operation for rectal cancer. This score can also be beneficial in reflecting the impact of symptoms on the quality of life of the patients [13, 24].

## 4.2 Validation of the LARS score

**4.2.1 Participants.** Due to the importance and simple application of the LARS score for evaluating LAR syndrome after sphincter-preserving operations, it has been validated in several languages. In 2012, the first version was designed and validated in the Danish language with 961 participants. Later in 2014, the international version [18], including four languages

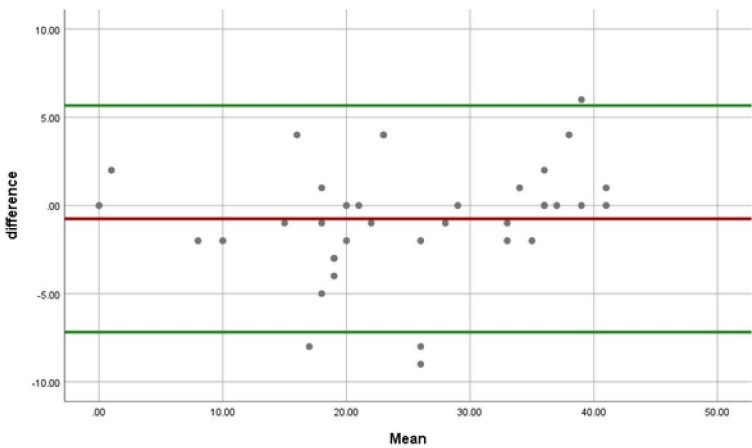

**Fig 3. Bland–Altman plot of the differences between LARS scores on the two tests.**

(Swedish, Spanish, German, and Danish languages) was validated with 801 participants. To validate the English version [14] in the same year, 451 patients were included. Chinese version [15] was validated in 2015 with 102 participants. In 2016, the Lithuanian version [16] was validated with 111 patients. Two versions of Dutch [17] and Japanese [19] were validated in 2018, with 165 and 149 participants. The mean age of participants in the previous studies was between 62.8 and 69.8 years [13–19]. We performed our analysis to validate the Persian version of the LARS score with 101 participants with a mean age of 57.2 years.

**4.2.2 Convergent validity, sensitivity, and specificity.**   In previously validated questionnaires [13–19], the ratio of patients with a perfect, moderate, and no fit between the QoL group and the LARS score category was reported as 41.8%-62.21%, 31.94%-49.7%, and 4.5%-8.5% respectively. Our results showed 82.1% for perfect fit, 16.9% for moderate, and 1% for no fit. In the Dutch and Danish versions, the mean LARS score was 12.9–20, 23.2–29.3, and 32.7 for those reporting no, minor, and major impact on QoL, respectively. Our data showed the same results for no (mean:18) and minor (mean: 26.7) impact on QoL; while the mean for major impact was higher (mean: 35.9). Moreover, all the studies, including our Persian version, have shown a statistically meaningful difference in the LARS score among the QoL groups. It means that the higher the LARS score, the more influence of symptoms on the QoL of the patient. The sensitivity of the previously validated version has been reported between 54.7% and 88.3%. Specificity has also been mentioned between 48.7% and 88.8%. Persian questionnaire also showed excellent sensitivity (97.3%) and specificity (78.7%) compared to previous ones.

**4.2.3 Discriminative validity.**   Previous studies have tried to see if their validated score can differentiate participants in terms of various variables. Discriminative validity in the Chinese version of the questionnaire [15] showed that the LARS score could differentiate between various groups of patients in terms of gender, history of previous radiotherapy, and length of the postoperative period (12 months). In the English validated score [14], preoperative radiotherapy, distance of the tumor from the anal verge, and extent of mesorectal excision were the variables showed statistically significant results. However, the length of the postoperative period was the only related variable in the Lithuanian study [16]. In the international version [18], three variables of gender, history of radiotherapy, and extent of mesorectal excision were significant in the discriminative validity. Age, preoperative radiotherapy and extent of mesorectal excision were the most effective items in the Dutch version [17]. In the Japanese version [19], discriminative validity showed that the LARS score could distinguish between the patients in terms of the tumor distance from the anal verge, time since surgery, and type of surgery. In our analysis, gender and T staging of the primary tumor were statistically significant variables.

**4.2.4 Test-retest reliability.**   Reliability investigation has been performed with 20 to 136 participants. We could perform the test-retest with the help of 37 participants. Various analytical methods have been utilized to check the reliability of the translated questionnaires. All of them proved to be highly reproducible in LAR patients. Paired t-tests have shown no significant difference between the LARS score on the first and second tests in English [15] and Japanese [19] translated scores. The international [18] and the Dutch [17] studies used the interclass correlation coefficient to analyze the reliability of their translated scores.

## 4.3 Strengths and limitations

This study validated the Persian (Farsi) version of the LARS score with a large number of participants compared to similar previous studies. However, studies with larger populations are recommended. Moreover, it was performed in a tertiary referral center with a group of patients

referred from the whole country showing its ability to be used for patients with different backgrounds. A group of eligible patients (261 out of 358) did not accept to participate in the study that was the main limitation of the study. The low response rate both in the initial phase of the study and also in the test-retest reliability are other possible sources of bias. The short time after stoma reversal (median 77 days in this study) might affect the LARS score as well because the bowel function might not be stable this early, and this may lead to some problems in the test-retest analysis.

## 5. Conclusion

The Persian translation of the LARS score demonstrated acceptable psychometric properties like other translated versions. It can be used as a valid tool for evaluating low anterior rectal syndromes and bowel function after sphincter-preserving operations, including low anterior resection for rectal cancer. The Persian score can also determine the severity of the symptoms and their influence on the quality of life of the patient. The Persian LARS score can also distinguish patients in terms of gender and T staging of the primary tumor.

## Supporting information

**S1 Data.**
(SAV)

## Author Contributions

**Conceptualization:** Mohammad Reza Keramati, Ali Abbaszadeh-Kasbi, Amir Keshvari, Seyed Mohsen Ahmadi-Tafti, Behnam Behboudi, Alireza Kazemeini, Mohammad Sadegh Fazeli.

**Data curation:** Mohammad Reza Keramati, Ali Abbaszadeh-Kasbi, Seyed Mohsen Ahmadi-Tafti, Behnam Behboudi, Alireza Kazemeini, Mohammad Sadegh Fazeli.

**Formal analysis:** Mohammad Reza Keramati, Ali Abbaszadeh-Kasbi, Seyed Mohsen Ahmadi-Tafti, Alireza Kazemeini.

**Investigation:** Mohammad Reza Keramati, Ali Abbaszadeh-Kasbi, Amir Keshvari, Seyed Mohsen Ahmadi-Tafti, Alireza Kazemeini, Mohammad Sadegh Fazeli.

**Methodology:** Mohammad Reza Keramati, Ali Abbaszadeh-Kasbi, Amir Keshvari, Seyed Mohsen Ahmadi-Tafti, Behnam Behboudi, Mohammad Sadegh Fazeli.

**Project administration:** Mohammad Reza Keramati, Ali Abbaszadeh-Kasbi, Amir Keshvari, Mohammad Sadegh Fazeli.

**Resources:** Mohammad Reza Keramati, Ali Abbaszadeh-Kasbi, Amir Keshvari, Behnam Behboudi, Alireza Kazemeini.

**Supervision:** Mohammad Reza Keramati, Ali Abbaszadeh-Kasbi, Amir Keshvari, Behnam Behboudi, Mohammad Sadegh Fazeli.

**Validation:** Mohammad Reza Keramati, Ali Abbaszadeh-Kasbi, Amir Keshvari, Seyed Mohsen Ahmadi-Tafti, Behnam Behboudi, Alireza Kazemeini, Mohammad Sadegh Fazeli.

**Visualization:** Alireza Kazemeini.

**Writing – original draft:** Mohammad Reza Keramati, Ali Abbaszadeh-Kasbi, Amir Keshvari, Seyed Mohsen Ahmadi-Tafti, Behnam Behboudi, Alireza Kazemeini, Mohammad Sadegh Fazeli.

**Writing – review & editing:** Mohammad Reza Keramati, Ali Abbaszadeh-Kasbi, Amir Keshvari.

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
