## [Decision Letter · Decision Letter 0]

2 Dec 2020

PONE-D-20-28626

Translation, Validation and Psychometric Evaluation of Farsi version of the Low Anterior Resection Syndrome Score

PLOS ONE

Dear Dr. Keramati,

Thank you for submitting your manuscript to PLOS ONE. After careful consideration, we feel that it has merit but does not fully meet PLOS ONE’s publication criteria as it currently stands. Therefore, we invite you to submit a revised version of the manuscript that addresses the points raised during the review process.

The manuscript and the reviewers’ comments were carefully evaluated. The manuscript was appreciated by the Reviewers. Nevertheless, as suggested, the manuscript requires improvements before to be considered for publication. Further suggested revisions are in detail reported in the Reviewers’ comments.

We look forward to receiving your revised manuscript.

Kind regards,

Simone Garzon

Academic Editor

PLOS ONE

Journal Requirements:

2. Thank you for stating in the text of your manuscript that you acquired permission from the original authors to translate the LARS score into Farsi. Please also add this information to your ethics statement in the online submission form.

3. In your Methods section, please provide additional information about the participant recruitment Please ensure you have provided sufficient details to replicate the analyses such as:

a) the recruitment date range (month and year),

b) a statement as to whether your sample can be considered representative of a larger population, and

c) a description of how participants were recruited.

Reviewers' comments:

Reviewer's Responses to Questions

**Comments to the Author**

1. Is the manuscript technically sound, and do the data support the conclusions?

Reviewer #1: Partly

Reviewer #2: Partly

Reviewer #3: Yes

2. Has the statistical analysis been performed appropriately and rigorously? 

Reviewer #1: No

Reviewer #2: No

Reviewer #3: Yes

3. Have the authors made all data underlying the findings in their manuscript fully available?

Reviewer #1: No

Reviewer #2: Yes

Reviewer #3: Yes

4. Is the manuscript presented in an intelligible fashion and written in standard English?

Reviewer #1: Yes

Reviewer #2: Yes

Reviewer #3: Yes

5. Review Comments to the Author

Reviewer #1: PONE-D-20-28626

Translation, Validation and Psychometric Evaluation of Farsi version of the Low Anterior Resection Syndrome Score

Thank you for the opportunity to review the manuscript. I reviewed this manuscript carefully with a great interest. I respectfully provided my comments below

1: It is recommended to change the title of the manuscript

Psychometric characteristics of the Persian version of the Low Anterior Resection Syndrome Score (LARS-P)

2: The abstract needs major editing.

3: In validity, explain face and content validity

4: In methods convergent and discriminative validity of the LARS score explain in two separate sections.

5: The standard time interval for the test re test is at least two weeks. Based on what reference did you consider this time period to be one week?

6: For test–retest reliability you must use Bland–Altman plot of the differences between LARS scores on the first and second tests.

Reviewer #2: The authors present the translation and psychometric validation of the e Low Anterior Resection Syndrome (LARS) score in Farsi. The sample size of 101 patients is sufficient for psychometric analyses.

Abstract

The abstract is punctual and well written.

Introduction

- Please provide reference for the second sentence in lines 57 and 58.

- Pleas change the last sentence of the introduction 'Herein, we would like to translate and validate the LARS score Farsi questionnaire and to measure its psychometric properties.' to clearly present an aim for the study in a separate paragraph. 'The aim of the present study was to translate the LARS score into Farsi and to test its psychometric properties.

Otherwise the introduction is well orchestrated.

Methods

Please provide reference for guidelines to which the translation process adhered. The most widely used guidelines are those of Beaton et al. and Wild et al. (ISPOR).

The authors should calculate reproducibility (test-retest reliability) values using the intraclass correlation coefficient (ICC), standard error of measurement (SEM) and if possible coefficient of repeatability (CR). This is the first time I have come across using the alpha for assessing test-retest reliability. If the authors prefer and insist in using the Cronbach's alpha coefficient (internal consistency) in test-retest reliability assessment, further clarification of analysis process and due references for the method should be provided. I could not find any suitable reference with a quick search for this method. Cronbach alpha provides an estimate of the internal consistency of the PROM and, to the best of my knowledge, does not indicate the stability or consistency of the test over time, which would be better estimated using the test-retest reliability strategy such as ICC, SEM and/or CR analyses. The authors can check this approach also from the COSMIN checklist (https://www.ncbi.nlm.nih.gov/pmc/articles/PMC2852520/)

Results

- Please provide a flow chart for patient selection.

- Please check the alpha in last row 208 according to my comment for the Methods section.

Discussion

- Please provide the main findings of the present study in the first paragraph of the Discussion section.

- Please provide a paragraph for strengths and weaknesses of the study

- The discussion is otherwise very good and provides sufficient comparison with previous studies.

Coclusion

This is fine.

Congratulations for this good paper. I will recommend minor to major revision before publication.

Reviewer #3: Review of Translation, Validation and Psychometric Evaluation of Farsi version of the Low Anterior Resection Syndrome Score

Thanks for giving me the opportunity to review this very relevant and well-written paper. The LARS score has proved to be a useful tool for measuring bowel dysfunction after sphincter preserving surgery for rectal cancer, and although it is relatively new, it has already been used in a high number of research projects, as well as in clinical practice, around the world. With the publication of this paper, it will be possible to conduct more research within this area in a population of Farsian speaking patients as well, which is very important. I hope the tool will be used to gain a deeper insight in the prevalence of bowel dysfunction and the impact on QoL after rectal cancer in Iran and other Farsian-speaking countries.

However, I have a few comments/questions/suggestions, which I hope the authors are willing to address:

Introduction:

(l 56) “…and the principal cause of cancer death in males and females, respectively”. I think that’s incorrect. Please check the results of the referenced paper.

Materials and methods:

(l 82-83) I believe it would be more correct to say that “The score-values were based on the item's impact on the qol of the patients.”

(l 129-30) “To evaluate the discriminative validity of the questionnaire, we tested the ability of the translated version of the LARS score to make a distinction between various groups of patients…” I would suggest adding “…who were expected to differ with regards to LARS” (and add some references which support this assumption)

(l 138) I don’t believe Cronbach’s alpha is appropriate for this tool measuring a SYNDROME consisting of five different symptoms, which are NOT necessarily strongly associated (but surely they sometimes are). If you are measuring a concept, such as anxiety (just an example) by means of summarizing several items, you would definitely require all of them to be closely related, and hence Cronbach´s alpha would be a relevant measure. If you still consider internal consistency relevant, please explain why and how we should interpret the result in detail.

Results:

In general, non-normally data should be presented as Median(IQR). Please check the distribution of your continuous variables, and make sure that you are using non-parametric tests whenever appropriate.

How many days between the two tests of the test-retest?

Table 1: “Stage of tumor" – is that UICC?

I think Table 2 is a bit confusing, since it is related to section 3.3. - Discriminative validity. The second part of the table (QoL) is not really relevant for any analyses of this validation paper, is it? You could consider dropping the QoL part, and move the table to section 3.3. (or drop Table 2 completely)

If “age” and/or “Distance of tumour from anal verge” are non-normally distributed, I think you should consider using the medians as cut off for the groups of in section 3.3. – Discriminative validity.

Discussion:

Please thoroughly discuss how these issues may impact the results/risk of bias:

Low response rate.

Short time after stoma reversal (33 days) (Bowel function might not be stable this early, and this might a problem in the test-retest – which requires a stable function).

Number of included patients.

Thanks.

6. PLOS authors have the option to publish the peer review history of their article (what does this mean?). If published, this will include your full peer review and any attached files.

Reviewer #1: **Yes: **Razieh Bandari

Reviewer #2: No

Reviewer #3: No

---

## [Author Response · Author response to Decision Letter 0]

10 Jan 2021

REVIEWER #1

Comment 1: It is recommended to change the title of the manuscript.

Answer: Thank you very much for your time. Title of the manuscript was revised according to your guidance. 

Comment 2: The abstract needs major editing.

Answer: Thanks. The abstract section was revised.

Comment 3: In validity, explain face and content validity.

Answer: Thanks. A subsection including face and content validity of the questionnaire was added to the manuscript.

Comment 4: In methods convergent and discriminative validity of the LARS score explain in two separate sections.

Answer: Thanks. The mentioned part in the methods section was divided into two subsections to describe convergent and discriminative validities separately.

Comment 5: The standard time interval for the test re test is at least two weeks. Based on what reference did you consider this time period to be one week?

Answer: Thank you very much for your comment. According to previous studies the recommended time for test-retest period is between one and two weeks. As mentioned in the manuscript, we did the retest after at least one week. Based on your guidance, the related references have been added to the related section of the methods section. 

Comment 6: For test–retest reliability you must use Bland–Altman plot of the differences between LARS scores on the first and second tests.

Answer: Thanks. The Bland-Altman plot of the test-retest was added to the manuscript. 

REVIEWER #2

Comment 1 – Introduction

Please provide reference for the second sentence in lines 57 and 58. 

Pleas change the last sentence of the introduction 'Herein, we would like to translate and validate the LARS score Farsi questionnaire and to measure its psychometric properties.' to clearly present an aim for the study in a separate paragraph. 'The aim of the present study was to translate the LARS score into Farsi and to test its psychometric properties. Otherwise the introduction is well orchestrated.

Answer: Thank you very much for your comments. According to the GLOBOCAN 2018( Reference number one) the sentences were revised. Last sentence of the introduction section was also revised according to your guidance. 

Comment 2 – Methods

Please provide reference for guidelines to which the translation process adhered. The most widely used guidelines are those of Beaton et al. and Wild et al. (ISPOR).

The authors should calculate reproducibility (test-retest reliability) values using the intraclass correlation coefficient (ICC), standard error of measurement (SEM) and if possible coefficient of repeatability (CR). This is the first time I have come across using the alpha for assessing test-retest reliability. If the authors prefer and insist in using the Cronbach's alpha coefficient (internal consistency) in test-retest reliability assessment, further clarification of analysis process and due references for the method should be provided. I could not find any suitable reference with a quick search for this method. Cronbach alpha provides an estimate of the internal consistency of the PROM and, to the best of my knowledge, does not indicate the stability or consistency of the test over time, which would be better estimated using the test-retest reliability strategy such as ICC, SEM and/or CR analyses. The authors can check this approach also from the COSMIN checklist (https://www.ncbi.nlm.nih.gov/pmc/articles/PMC2852520/)

Answer: Thank you very much. The reference for the translation process was added. According to your guidance, ICC, SEM and CR were calculated and added to both the method and the results sections of the manuscript.

Comment 3 – Results

Please provide a flow chart for patient selection. - Please check the alpha in last row 208 according to my comment for the Methods section.

Answer: Thanks. Flowchart for the patient selection has been added to manuscript as figure 1. According to your guidance, ICC, SEM and CR were calculated and added to both the method and the results sections of the manuscript.

Comment 4 - Discussion

- Please provide the main findings of the present study in the first paragraph of the Discussion section.

- Please provide a paragraph for strengths and weaknesses of the study

- The discussion is otherwise very good and provides sufficient comparison with previous studies.

Answer: Thank you very much. Main findings of the study have been added to the first paragraph of the discussion section. Strengths and limitations of the study has also been added to the end of discussion section. 

Reviewer #3

Comment 1 - Introduction:

(l 56) “…and the principal cause of cancer death in males and females, respectively”. I think that’s incorrect. Please check the results of the referenced paper.

Answer: Thank you very much for your comments: According to the GLOBOCAN 2018( Reference number one) the sentences were revised.

Comment 2 - Materials and methods:

(l 82-83) I believe it would be more correct to say that “The score-values were based on the item's impact on the qol of the patients.”

(l 129-30) “To evaluate the discriminative validity of the questionnaire, we tested the ability of the translated version of the LARS score to make a distinction between various groups of patients…” I would suggest adding “…who were expected to differ with regards to LARS” (and add some references which support this assumption)

(l 138) I don’t believe Cronbach’s alpha is appropriate for this tool measuring a SYNDROME consisting of five different symptoms, which are NOT necessarily strongly associated (but surely they sometimes are). If you are measuring a concept, such as anxiety (just an example) by means of summarizing several items, you would definitely require all of them to be closely related, and hence Cronbach´s alpha would be a relevant measure. If you still consider internal consistency relevant, please explain why and how we should interpret the result in detail.

Answer: Thank you very much. According to your guidance, the sentences in the first paragraph of the methods section and the convergent validity part of the methods section were revised and completed. Related references were also added. According to your guidance, intraclass correlation coefficient (ICC), standard error of measurement (SEM) and coefficient of repeatability (CR) were calculated and added to both the method and the results sections of the manuscript.

Comment 3 - Results:

In general, non-normally data should be presented as Median(IQR). Please check the distribution of your continuous variables, and make sure that you are using non-parametric tests whenever appropriate.

How many days between the two tests of the test-retest?

Table 1: “Stage of tumor" – is that UICC?

I think Table 2 is a bit confusing, since it is related to section 3.3. - Discriminative validity. The second part of the table (QoL) is not really relevant for any analyses of this validation paper, is it? You could consider dropping the QoL part, and move the table to section 3.3. (or drop Table 2 completely)

If “age” and/or “Distance of tumour from anal verge” are non-normally distributed, I think you should consider using the medians as cut off for the groups of in section 3.3. – Discriminative validity.

Answer: Thank you very much. Median was calculated for continuous variables and added to the manuscript. The second test was done at least one week after the first test. This has been mentioned in the manuscript. Tumor were staged according to the UICC TNM staging system that added to the manuscript. Table 2 was removed totally from the manuscript. As you mentioned regarding the variables age (median=57) and distance of tumor from anal verge (median=8) we used medians as cut off for the groups of in section 3.3.

Comment 4 - Discussion:

Please thoroughly discuss how these issues may impact the results/risk of bias:

Low response rate.

Short time after stoma reversal (33 days) (Bowel function might not be stable this early, and this might a problem in the test-retest – which requires a stable function).

Number of included patients.

Answer: Thank you very much. Regarding the risk of possible bias, each mentioned item has been discussed in the limitation part of the paper (last part of the discussion section). 

Journal Requirements:

Comment 1. Please ensure that your manuscript meets PLOS ONE's style requirements, including those for file naming.

Answer: Thank you very much. The manuscript was revised based on the journal’s guideline.

Comment 2. Thank you for stating in the text of your manuscript that you acquired permission from the original authors to translate the LARS score into Farsi. Please also add this information to your ethics statement in the online submission form.

Answer: Thanks. The sentence was added to the online ethics section. 

Comment 3. In your Methods section, please provide additional information about the participant recruitment. Please ensure you have provided sufficient details to replicate the analyses such as:

a) the recruitment date range (month and year),

b) a statement as to whether your sample can be considered representative of a larger population, and

c) a description of how participants were recruited.

Answer: Thank. Recruitment date and description of the recruitment are available in the participants part of the methods section. A sentence has been added to this section showing that the participants are representative of a larger population. 

Comment 4. We note that you have stated that you will provide repository information for your data at acceptance. Should your manuscript be accepted for publication, we will hold it until you provide the relevant accession numbers or DOIs necessary to access your data. If you wish to make changes to your Data Availability statement, please describe these changes in your cover letter and we will update your Data Availability statement to reflect the information you provide.

Answer: Thank you very much. Data of this study is completely available and we will be happy to provide the data upon acceptance of the manuscript. 

Comment 5. Your ethics statement should only appear in the Methods section of your manuscript. If your ethics statement is written in any section besides the Methods, please delete it from any other section.

Answer: Thanks. The statement regarding the ethics approval has been mentioned once in the methods section.

---

## [Decision Letter · Decision Letter 1]

29 Jan 2021

PONE-D-20-28626R1

Translation, Validation and Psychometric Evaluation of the Persian (Farsi) version of the Low Anterior Resection Syndrome Score (LARS-P)

PLOS ONE

Dear Dr. Keramati,

Thank you for submitting your manuscript to PLOS ONE. After careful consideration, we feel that it has merit but does not fully meet PLOS ONE’s publication criteria as it currently stands. Therefore, we invite you to submit a revised version of the manuscript that addresses the points raised during the review process.

The manuscript and the reviewers’ comments were carefully evaluated. The manuscript was appreciated by the Reviewers. However, minor revisions are required before final acceptance.

We look forward to receiving your revised manuscript.

Kind regards,

Simone Garzon

Academic Editor

PLOS ONE

Reviewers' comments:

Reviewer's Responses to Questions

**Comments to the Author**

1. If the authors have adequately addressed your comments raised in a previous round of review and you feel that this manuscript is now acceptable for publication, you may indicate that here to bypass the “Comments to the Author” section, enter your conflict of interest statement in the “Confidential to Editor” section, and submit your "Accept" recommendation.

Reviewer #1: All comments have been addressed

Reviewer #2: All comments have been addressed

Reviewer #3: (No Response)

2. Is the manuscript technically sound, and do the data support the conclusions?

Reviewer #1: Yes

Reviewer #2: Yes

Reviewer #3: Yes

3. Has the statistical analysis been performed appropriately and rigorously? 

Reviewer #1: Yes

Reviewer #2: Yes

Reviewer #3: Yes

4. Have the authors made all data underlying the findings in their manuscript fully available?

Reviewer #1: Yes

Reviewer #2: No

Reviewer #3: Yes

5. Is the manuscript presented in an intelligible fashion and written in standard English?

Reviewer #1: Yes

Reviewer #2: Yes

Reviewer #3: Yes

6. Review Comments to the Author

Reviewer #1: (No Response)

Reviewer #2: I recommended publication as all issues raised by the Reviewers are accounted for. I have no further comments or requirements.

Reviewer #3: line 295 (Discussion):" Cronbach’s alpha displayed good internal consistency and reliability in our study". I believe you took this analysis out of the manuscript, hence it should not be mentioned in the discussion.

I recommend to show not just the minimum days between the two tests in the test-retest, but also the maximum number of days.

7. PLOS authors have the option to publish the peer review history of their article (what does this mean?). If published, this will include your full peer review and any attached files.

Reviewer #1: No

Reviewer #2: No

Reviewer #3: No

---

## [Author Response · Author response to Decision Letter 1]

29 Jan 2021

Dear Editor in Chief of the PLOS Journal,

Thank you very much for considering our manuscript and for the valuable comments of the reviewers. We revised the manuscript based on the comments. All the changes are highlighted within the manuscript. Answers to the questions of the reviewers are in below.

Again, thanks for considering our manuscript. I really appreciate it if you may inform me if further revision is needed for our manuscript.

Sincerely,

M.R.Keramati 

REVIEWER #1

Comment: All comments have been addressed.

Answer: Thank you very much for your time and valuable comments.

REVIEWER #2

Comment: I recommended publication as all issues raised by the Reviewers are accounted for. I have no further comments or requirements.

Answer: Thank you very much for your time and valuable comments.

Reviewer #3

Comment 1: line 295 (Discussion):" Cronbach’s alpha displayed good internal consistency and reliability in our study". I believe you took this analysis out of the manuscript, hence it should not be mentioned in the discussion.

Answer: Thank you very much for your time and valuable comments. According to your comment, the analysis (regarding the Cronbach’s alpha) was removed from the mentioned part of the discussion section.

Comment 2: I recommend to show not just the minimum days between the two tests in the test-retest, but also the maximum number of days.

Answer: Thank you very much. In line-141 of the manuscript (Reliability part of the Methods Sections) the maximum number of days was added.

---

## [Decision Letter · Decision Letter 2]

1 Feb 2021

Translation, Validation and Psychometric Evaluation of the Persian (Farsi) version of the Low Anterior Resection Syndrome Score (LARS-P)

PONE-D-20-28626R2

Dear Dr. Keramati,

We’re pleased to inform you that your manuscript has been judged scientifically suitable for publication and will be formally accepted for publication once it meets all outstanding technical requirements.

Kind regards,

Simone Garzon

Academic Editor

PLOS ONE

Additional Editor Comments (optional):

Reviewers' comments:

Reviewer's Responses to Questions

**Comments to the Author**

1. If the authors have adequately addressed your comments raised in a previous round of review and you feel that this manuscript is now acceptable for publication, you may indicate that here to bypass the “Comments to the Author” section, enter your conflict of interest statement in the “Confidential to Editor” section, and submit your "Accept" recommendation.

Reviewer #3: All comments have been addressed

2. Is the manuscript technically sound, and do the data support the conclusions?

Reviewer #3: Yes

3. Has the statistical analysis been performed appropriately and rigorously? 

Reviewer #3: Yes

4. Have the authors made all data underlying the findings in their manuscript fully available?

Reviewer #3: Yes

5. Is the manuscript presented in an intelligible fashion and written in standard English?

Reviewer #3: Yes

6. Review Comments to the Author

Reviewer #3: (No Response)

7. PLOS authors have the option to publish the peer review history of their article (what does this mean?). If published, this will include your full peer review and any attached files.

Reviewer #3: No

---

## [Editor Report · Acceptance letter]

17 Feb 2021

PONE-D-20-28626R2 

Translation, Validation and Psychometric Evaluation of the Persian (Farsi) version of the Low Anterior Resection Syndrome Score (LARS-P) 

Dear Dr. Keramati:

I'm pleased to inform you that your manuscript has been deemed suitable for publication in PLOS ONE. Congratulations! Your manuscript is now with our production department. 

Kind regards, 

on behalf of

Dr. Simone Garzon 

Academic Editor

PLOS ONE